# Molecular Characterization and Demographic Study on Infectious Bursal Disease Virus in Faisalabad District

Sanaullah Sajid[1], Sajjad ur Rahman[1]*, Mashkoor Mohsin Gilani[1], Zia ud Din Sindhu[2], Manel Ben Ali[3], Amor Hedfi[3], Mohammed Almalki[3], Shahid Mahmood[1]

1 Institute of Microbiology, University of Agriculture, Faisalabad, Pakistan, 2 Department of Parasitology, University of Agriculture, Faisalabad, Pakistan, 3 Department of Biology, College of Sciences, Taif University, Taif, Saudi Arabia

* Sajjadur@gmail.com

**Data Availability Statement:** All virus isolate files are available from the NCBI database (MW483681, MW483682, MW483683, MW483684).

## Abstract

The re-emergence of virulent strains of the Infectious Bursal Disease Virus (IBDV) leads to significant economic losses of poultry industry in Pakistan during last few years. This disease causes the infection of bursa, which leads to major immune losses. A total number of 30 samples from five IBD outbreaks during the period of 2019–20 were collected from different areas of Faisalabad district, Pakistan and assayed by targeting the IBD virus VP2 region through RT-PCR. Among all the outbreaks, almost 80% of poultry birds were found positive for the IBDV. The bursa tissues were collected from the infected birds and histopathological examination of samples revealed severe lymphocytic depletion, infiltration of inflammatory cells, and necrosis of the bursa of Fabricius (BF). Positive samples were subjected to re-isolation and molecular characterization of IBDV. The Pakistan IBDV genes were subjected to DNA sequencing to determine the virus nucleotide sequences. The sequences of 100 Serotype-I IBDVs showing nearest homology were compared and identified with the study sequence. The construction of the phylogenetic tree for nucleotide sequences was accomplished by the neighbor-joining method in MEGA-6 with reference strains. The VP2 segment reassortment of IBDVs carrying segment A were identified as one important type of circulating strains in Pakistan. The findings indicated the molecular features of the Pakistan IBDV strains playing a role in the evolution of new strains of the virus, which will contribute to the vaccine selection and effective prevention of the disease.

## Introduction

Gumboro is a highly contagious disease of fowl caused by the infectious bursal disease virus (IBDV). The virus belongs to the Avibirnaviridae group of Birnaviridae family. IBD virus is a non-enveloped dsRNA virus with a genome having two segments A and B with a diameter of 55–60 nm. Two serotypes of this virus have been recognized, of which only Serotype-I is known to cause disease in birds [1]. Serotype-II infects turkeys and several avian species, but

**Funding:** The authors extend their appreciation to the deanship of scientific research for funding this article by Taif University Research Supporting Project number (TURSP-2020/225), Taif University, Taif, Saudi Arabia.The funders had role in data analysis and preparing the manuscript.

**Competing interests:** No authors have competing interests.

no disease has been reported due to it. Segment A is self-cleaved to yield protease and structural proteins pVP2 and VP3. The pVP2 encodes the VP2 as a major capsid protein of the virus. A wide variety of pathotypes of serotype-I exists in nature that has been classified as clinical virulent, subclinical virulent and very virulent groups [2, 3]. Despite the efforts of vaccination, outbreaks of IBDV are still reported worldwide as a significant immunosuppressive disease of fowl. The re-assortment of IBDV genome, genomic recombination, and frequent genetic mutations potentially alter antigenicity and increase the virulence which renders the vaccines least effective. Since the virus is resistant to heat inactivation, chemicals, and highly contagious so, eradication is not practical on infected farms [4].

The economic losses due to IBD experienced by the poultry industry are not only the result of mortality and morbidity but the dramatic fall in the overall performance of the flock. Bursa of Fabricius is the target organ of IBDV that replicates in the B-lymphocytes resulting in humoral suppression due to depletion of B cells [5]. Thus, increased susceptibility to the multitude of opportunistic pathogens due to irreversible immune suppression in the IBD infected birds. IBD results in the economic impacts on the layer and broiler chicken industry that is estimated to be 3.9 million kilograms of meat per year having $14 million market value [6, 7]. Infectious Bursal Disease Virus is present in clinical and subclinical forms and birds recovered from the virus remains carrier. There is no treatment for this disease, but vaccination and biosecurity [8, 9]. The two major antigenic groups within serotype-I are commonly called classical and variant, but antigenic drift has contributed to the formation of several subtypes within these groups. The antigenic phenotype of IBDV is determined by the hypervariable region of VP2 (hvVP2) specifically, by amino acids located at the apex of loop structures designated $P_{BC}$, $P_{DE}$, $P_{FG}$, and $P_{HI}$. Even single point mutations in these regions have been found to contribute to antigenic drift in IBDV, which can render currently available IBD vaccines ineffective. The phylogenetic studies suggested that the expansion of the virus was initiated where an endemic segment A was reasserted with segment B of unknown origin [10, 11].

Our study aimed to identify IBDV strains that continue to affect and cause disease in commercial chicken flocks. Although pathogenicity is important concerning the severity of the disease and degree of immune suppression, the current study focused on mutations located in the hvVP2.

## Materials and methods

### Field samples

Symptomatic BF samples were submitted from different commercial farms in Faisalabad district. During the year 2019 to 2020, a total of 30 BF were obtained from five IBD-suspected outbreaks in Faisalabad, with 10–18% mortality observed at these farms (Table 1). The infected birds were manifested by atrophy and inflammation of BF and the feces stained feathers around the vent. The BF samples were transported to the Molecular Research Laboratory, Institute of Microbiology, University of Agriculture Faisalabad, Pakistan for molecular diagnosis and characterization after approval from Institutional Biosafety Committee (IBC).

### Histopathology

Histopathology of tissues was performed as described by [12]. Briefly, Bursa tissues were fixed in 10% neutral buffered formalin solution for seven days. The samples were washed with water to remove the fixative and dehydrated with increasing concentration of ethanol, followed by clearance in xylene. Tissues were embedded in paraffin using steel molds and sectioned to 3 μm thickness with a microtome. Using Mayer's egg albumin as an adhesive material, tissues

**Table 1. Field samples collected from flocks with signs of disease.**

| Sample ID | Place and Year | Flock size | Age (days) | Category | Vaccination status | Reported Pathogenicity |
|---|---|---|---|---|---|---|
| IOM-2001 | F-2019 | 5500 | 30 DO | Broiler | V | Low |
| IOM-2002 | F-2019 | 6500 | 22 DO | Broiler | NR | Low |
| IOM-2003 | F-2019 | 10000 | 35 DO | Layer | NV | High |
| IOM-2004 | F-2019 | 10000 | 28 DO | Broiler | NV | NR |
| IOM-2005 | F-2019 | 1750 | 30 DO | Layer | V | Low |
| IOM-2006 | F-2019 | 5700 | 49 DO | Layer | V | High |
| IOM-2007 | F-2019 | 1200 | 42 DO | Layer | NV | High |
| IOM-2008 | F-2019 | 20000 | 42 DO | Layer | NV | NR |
| IOM-2009 | F-2019 | 1000 | 29 DO | Broiler | NR | Low |
| IOM-2010 | F-2019 | 15700 | 27 DO | Broiler | V | Low |
| IOM-2011 | F-2019 | 10000 | 36 DO | Layer | V | NR |
| IOM-2012 | F-2019 | 1250 | 28 DO | Broiler | V | Low |
| IOM-2013 | F-2020 | 5000 | 27 DO | Broiler | V | NR |
| IOM-2014 | F-2020 | 5200 | 36 DO | Broiler | NV | High |
| IOM-2015 | F-2020 | 6000 | 30 DO | Broiler | V | Low |
| IOM-2016 | F-2020 | 1000 | NR | NR | NV | High |
| IOM-2017 | F-2020 | 6500 | 35 DO | Broiler | V | Low |
| IOM-2018 | F-2020 | 5500 | 34 DO | Broiler | NR | Low |
| IOM-2019 | F-2020 | 1250 | 28 DO | Broiler | V | Low |
| IOM-2021 | F-2020 | 5800 | NR | NR | V | High |
| IOM-2022 | F-2020 | 10000 | 25 DO | Broiler | V | High |
| IOM-2023 | F-2020 | 1200 | 20 DO | Broiler | NR | NR |
| IOM-2024 | F-2020 | 10000 | NR | NR | V | Low |
| IOM-2025 | F-2020 | 30000 | 28 DO | Broiler | V | Low |
| IOM-2026 | F-2020 | 500 | 25 DO | Broiler | V | NR |
| IOM-2027 | F-2020 | 570 | 31 DO | Broiler | V | High |
| IOM-2028 | F-2020 | 900 | 30 DO | Broiler | V | High |
| IOM-2029 | F-2020 | 600 | NR | NR | V | NR |
| IOM-2030 | F-2020 | 1000 | 23 DO | Broiler | V | Low |

F: Faisalabad, DO: day-old, NR: not reported, V: vaccinated, NV: Non vaccinated

were mounted on glass slides and dried at 37˚C for 24 hours. After incubation for 30 minutes at 45˚C, tissues were stained with Haematoxylin and eosin stains [13, 14].

## Electron microscopy

A small fraction of virus suspension was taken after trituration of bursa centrifuged at 8000 rpm for five minutes. The suspension was treated with 0.05 M Tris-HCl having 8.0 pH. After staining with 2% sodium phosphotungstate, virus particles were adsorbed to the formvar carbon-coated grid. Specimen grid was examined in an electron microscope operating at 100 kV [15].

## RNA isolation

The bursae of Fabricius were homogenized in mortar and pestled with phosphate buffered saline supplemented with Kanamycin (700 μg/ml). Samples were clarified by centrifugation at 8000 rpm for five minutes to eliminate the cell debris. Total RNA was extracted using 200 μl of

the supernatant using GeneJet viral purification kit (Thermo Fisher) according to the manufacturer's protocol [16, 17].

## Quantification of complementary DNA

Complementary DNA was quantified in the analytical laboratory of US-Pak Center for Advanced Studies, University of Agriculture Faisalabad, Pakistan using the Nano-Drop quantification method. The elution buffer was run as a blank and one μl of viral cDNA samples were placed to record the readings [18].

## RT-PCR

SuperScript™ reverse transcriptase kit (Invitrogen, USA) was used for RT-PCR reaction. Total RNA template was mixed in a 20μl reaction containing 50 ng of random hexamer primer, 10 mM dNTP mix, 4 μl of 5X First-Strand buffer, 1 μl of 0.1 M DDT, and 200 units of Super-Script™ RT. The IBDV hypervariable VP2 region was detected by using primer pair BG-VP2 designed using LaserGene software. The primer pair BG-VP2 produces a DNA amplicon of 699 base pairs. The visualization of the PCR product was done by electrophoresis on a 1.5% agarose gel stained with Ethidium Bromide [19, 20].

## Sequence analysis of the hypervariable region of IBDV-VP2 gene

The purification of the amplified PCR products of VP2 (hypervariable) region were done by PCR purification kit (QIAGEN, USA) according to the protocol given by the manufacturer. Purified VP2 DNA was subjected to DNA sequencing at Advance Bioscience Internationals to determine the virus nucleotide sequences. The sequences were then deposited in the Gen-Bank/NCBI database and the deduced amino acid and nucleotide sequences of IBDV were aligned using the CLUSTAL-W method and subjected to BLAST searches [https://blast.ncbi.nlm.nih.gov/Blast.cgi] to determine the identity with other strains [21]. A total 100 sequences of Serotype-I IBDVs showing the nearest homology were compared and identified with the study sequence. Representative strains were used to construct the phylogenetic tree for nucleotide sequences. The bootstrap method was used to estimate the topological accuracy of the phylogenetic tree, with 1000 replicates and the neighbor-joining method was used for the determination of inference of phylogenetic relationship in MEGA-6 [22, 23].

## Phylogenetic analysis of VP2 hypervariable region

Nucleotide sequence data were analyzed by MegaAlign, BioEdit, and Lasergene DNASTAR software's using the hypervariable region of the VP2 gene. Molecular and phylogenetic analysis was conducted using Mega software (version 6.0) employing the neighbor-joining method with Kimura three-parameter evolutionary model. The phylogenetic analysis relies on the arrangement of VP2 nucleotide sequences covering the hypervariable region at the nucleotide 845–1126. The deduced amino acid sequences from 210 to 391 were investigated for genetic markers. The sequence of isolates incorporated in the Table 2 were grouped into seven Geno groups. Sequenced data from Asia and other regions were selected for the comprehensive studies to evaluate the genetic information based on the neighbor-joining method. The sequence profile of deduced amino acid in the hydrophilic regions was evaluated to explore the mutations at the amino acid level.

**Table 2. Description of genetic markets in different geno-groups.**

| Geno-Group | 222 [a] | 242 [a] | 256 [a] | 279 [a] | 294 [a] | 299 [a] |
|---|---|---|---|---|---|---|
| G-1 | A/P/S | V/2 | V/I/A | D/N | I/L | N |
| G-2 | T/Q | V | V | N | L | N/S |
| G-3 | A/S | I | I | D/N | I | S/N |
| G-4 | S | V | V | N | L | H/S/N |
| G-5 | T | V | V | N | L | N |
| G-6 | Q | V | K/I | D | L | S |
| G-7 | P | V | V | G | L | S |

a: Amino acid positions in VP2 region of IDB virus

## Results

### Histopathological examination

Histopathological examination of the bursae showed characteristic lesions attributed to the IBDV infection. Acute and chronic lesions were identified in 30 infected bursae with fibrin deposition and depletion of lymphoid follicles, necrosis, infiltration of heterophilic inflammatory cells, severe atrophy with the presence of vacuoles in the medullary and cortical region. During infection, the necrosis of lymphocytes resulted in karyorrhectic debris followed by characterized edema, hyperemia, and reticuloendothelial cell hyperplasia. The inflammation diminished on the fifth day of the infection by removal of debris through phagocytosis and the appearance of cystic cavities in the medullary area of lymphoid follicles (Figs 1 and 2).

### Electron microscopy

The determination of the icosahedral structure of the IBDV was confirmed from the images obtained by the transmission electron microscope. The reconstruction of IBDV revealed the

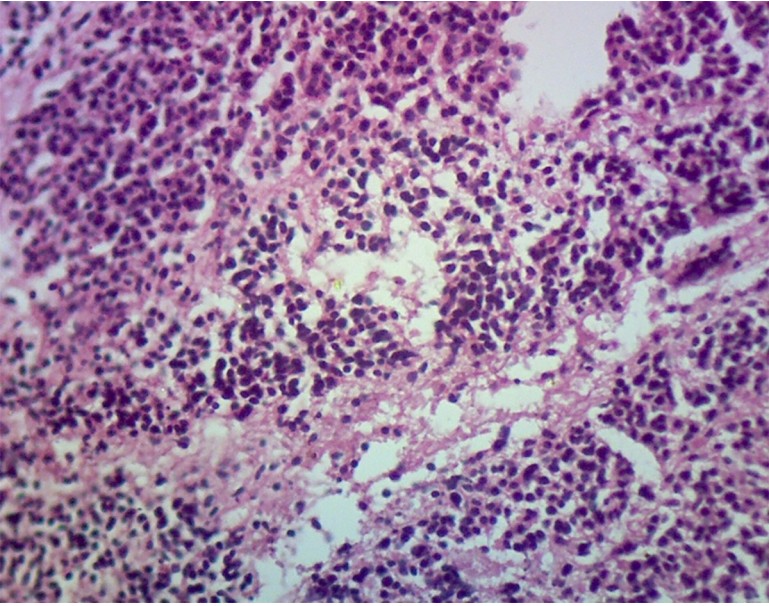

**Fig 1. Histopathological section of the BF of chicken with infiltration of inflammatory cells, desquamation of surface epithelium (40X magnification, H&E staining).**

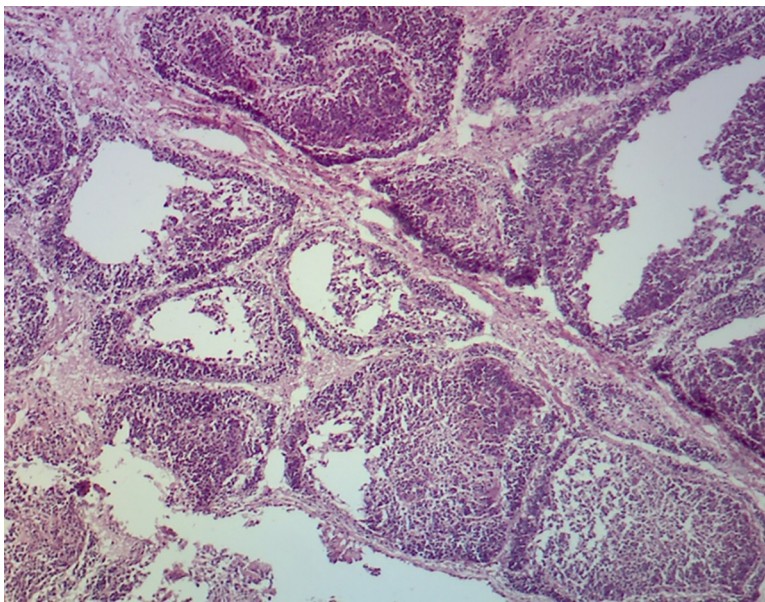

**Fig 2. Histopathological section of BF of chicken with lymphocytic depletion with marked atrophy of lymphoid follicles and thickening of interfollicular space by severe fibrosis (40X magnification, H&E staining).**

icosahedral shape of virus particles with pentameric protrusions on each fivefold vertex. The five to fivefold distance measures 750 angstroms, in which the VP2 protein of IBD virus appeared to be organized in 260 trimers in fivefold vertices, making the edges and the face of the icosahedron (Fig 3).

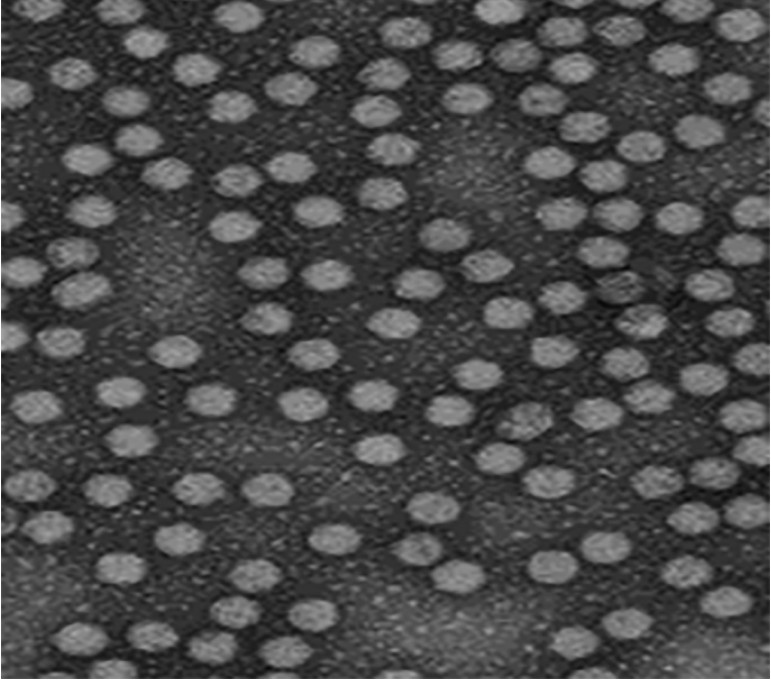

**Fig 3. Transmission electron microscopy IBDV after negative staining with 2% sodium phosphotungstate.**

**Table 3. Infectious bursal disease virus strains isolated in the study.**

| No. | Strain | Origin | Collection Date | Host | Accession no. |
|---|---|---|---|---|---|
| 1 | IOMF2019 | Faisalabad, Punjab | 2019 | Layer | MW483681 |
| 2 | IOMF2019 | Faisalabad, Punjab | 2019 | Broiler | MW483682 |
| 3 | IOMF2020 | Faisalabad, Punjab | 2020 | Broiler | MW483683 |
| 4 | IOMF2020 | Faisalabad, Punjab | 2020 | Broiler | MW483684 |

## Molecular detection of IBDV VP2 gene

RT-PCR showed that 13% (4/30) of bursa tissues were positive for IBDV. A partial region in segment A (reg VP2) was amplified for sequencing and comparative analysis of isolates. The partial VP2 region was successfully amplified in the present study and added to the analysis. All the four Pakistan strains were submitted to GenBank, and the accession numbers are listed in Table 3.

## Sequence analysis of VP2-HVR of geno-groups

For the detailed analysis, all the sequences were aligned for nucleotide position at 628–1173 and amino acid position at 210–391. Based on constructed phylogenetic tree and similarities, the data displayed into seven genogroups. genogroup 1, comprising 15 globally reported and diversified isolates in which an amino acid position 222, over 80% of the isolates had Proline (P) and shared S/A in the variable region. Most isolates had Valine (V) at position 242. At position 256, the positioned amino acid Valine (V) was contributed to 88% along with Alanine (A) and Isoleucine (I). At position 279 in the minor hydrophilic region, the isolates had 12% Asparagine (N) and Aspartate (D), while Leucine (L) was positioned at 294.

In genogroup 2, the amino acids at position 222 in the conserved regions were Threonine (T), 256V, 296L, 242V, 299N, and 279N. The reference isolates in this group were limited to non-Asian countries. Previously, the classification of the isolates in this group was antigenic variants and confused with classical recombinants of genogroup 5. For the first time, these were clearly categorized into a separate group by Jackwood in December 2011 that presented with a universal prevalence of genogroup 3 with typical hallmarks similar at 242I, 222A, 256I, and 299 S (Serine). Three unique changes at amino acid S317R, D323N, and S315T were also reported and the isolates from the Asian border sharing countries had exhibited a heterologous record in the hypervariable regions of the IBDV. In genogroup 4, the amino acid exhibited H/S/Histidine (H) at genetic marker 299, while there were some common amino acids at position 294L, 222S, 279N, and 256V. This group had diversified amino acid profiles but showed similarities to genogroup 5 and prevalent all over the world. Genogroup 5 shared the same genetic hallmark including 294L, 256V, 222T, and 299N. Genogroup 6 is only prevailing outside Asia especially in Russia and Italy, while all the isolates in genogroup 7 shared the same amino acid Proline (P) at position 222 and this group is prevalent outside Asia. All the isolates of this group have shown 100% amino acid similarity.

The phylogenetic studies revealed that genogroups 1, 3, and 4 shared the genetic information universally, but the other groups were preserved in the specific regions. Classical strains of IBD virus were segregated in genogroup 1, while very virulent IBDV strains were grouped in genogroup 3 and these two groups consist of the most existing strains of the virus. The antigen variant strains of the virus are responsible for most economic losses in the poultry sector belonged to genogroup 2. The distinctive class of IBD virus (dIBDV) previously prevalent outside Asia was classified in genogroup 4. In the current study, one sample matched with the group 4 lineage needs further investigation. Recombinant IBDV classified in genogroup 5 shared the variant and classical group amino acid profile. The amino acid and genetic profiling

of genogroup 6 were reported from Italy and it was later on reported in Russia. The isolates of genogroup 7 were of Australian origin and few were reported in Russia (Figs 4 and 5).

## Discussion

Among the viral diseases of the poultry flocks, IBDV is the second foremost disease after New-castle disease. Due to the immunosuppressive nature of the disease, it continuously adds to the economic losses in the poultry sector due to compromised cellular and humoral responses after the destruction of B cells resulting in the lower feed conversion ratio. The only way to control IBD is vaccination however, the outbreaks of this disease are also documented in the vaccinated flocks due to the introduction of new virus strains [1] The possible reason could be the mutation (amino acid changes) that may alter the protection level of the chickens. The genogroups 1, 2 and 3 accommodate the large population of the IBDVs while the classical strains fall in gen-ogroup 1. The vaccination of the chickens is the only way to mitigate the losses caused by the classical IBD, while the frequent vaccination at different time intervals in the rearing flock could be a possible cause of mutation in the IBDVs. The quantity, quality, and procedure of vaccina-tion are also questionable. Despite the variation in the genetic hallmark of Pak-Asian isolates, the position of amino acids at N299 was found conserved in this genogroup [8].

Genogroup 2 shared the amino acid at position 279N, 299N, 318D, 249K, 256V, 242V, and 222T. The most economic losses in poultry industry were reported due to variant strains that lead to misdiagnosis and no pathognomonic lesions resulted from the secondary infections caused by immunosuppression. Genogroup 3 was reported to be responsible for the global outbreaks of IBD. The four-loop structures in the hypervariable region of VP2 (PDE, PHI, PBC, PFG) were linked to the modification in virulence of the IBDV [10]. In genogroup 4, the amino acid shared the common positions at 296F, 256V, 242V, 279N, and 289P. The sequence analysis showed the 92% similarity between the dIBDV and vaccinal strain of IBDV (Winter-field) that predicts the spread of these strains in Asian and non-Asian countries [24].

Genogroup 5 has common genetic markers with genogroup 2, but this group is restricted to countries other than Asia ones. Two substitutions at A321P and S317K were observed in the major hydrophilic region. At the antigen epitopes of a single IBDV, the presence of classical and variant sequences provides solid ground to genogroup 5. The IBDV in genogroup 6 was found to be circulating in the Italian region, but its sequenced data shared the worldwide dis-tribution of the virus [4]. The important gene markers of these isolates included 256K and 253E but the alteration of amino acid at positions 222Q, 220H, and 321A in major hydrophilic regions A and B classifies these isolates in an independent genogroup 6. In genogroup 7, mostly the Australian isolates were grouped, sharing 100% similarity due to conserved genetic markers and presence in the specific regions [2].

In the present scenario, trade and the vaccinal strains of the IBD virus might be the reasons for the global emergence of new strains with a difference in their pathogenicity. In Pakistan, 16 isolates were submitted in GenBank before the present study. In this region, classical strains of the IBDV before the 1980s were reported to be the dominant strains of the virus. The mutation at position 222 from Proline (P) to Throline (T) was considered in the shifting of classical to variant genogroup, which may be reflected from worldwide use of vaccine strains due to its immunization in rearing flocks.

## Conclusion

In the current study, the Infectious Bursal Disease Virus isolates were found to be present in genogroup 1, genogroup 4, and genogroup 5. All three genogroups are prevalent globally while the re-assorted and mutated isolated were confined to the specific regions of the world.

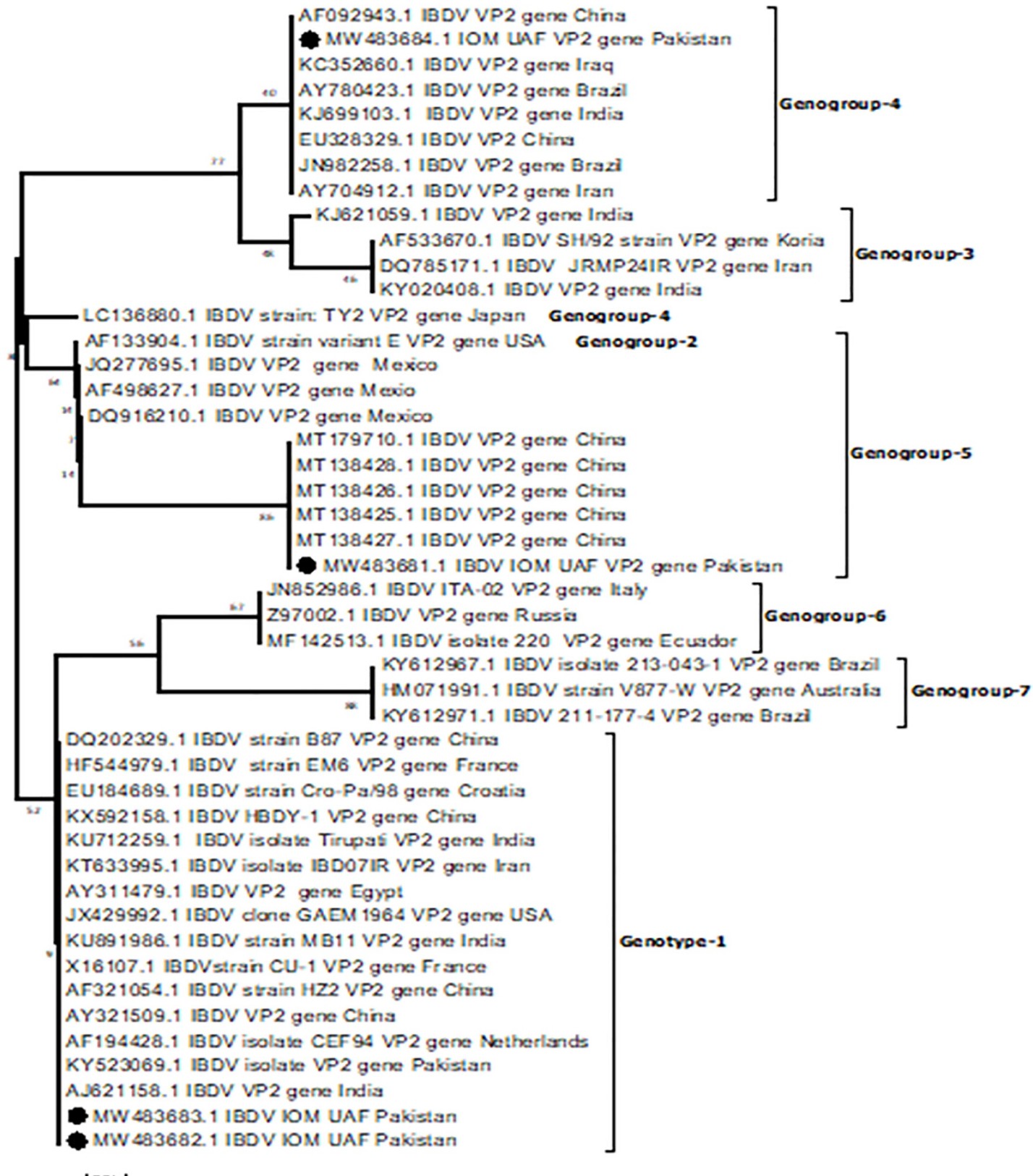

**Fig 4. Phylogenetic consensus tree based on the hypervariable domain of VP2 gene of infectious bursal disease virus, designed by the neighbor-joining method using Clustal W alignment of the nucleotides created in Mega-6.**

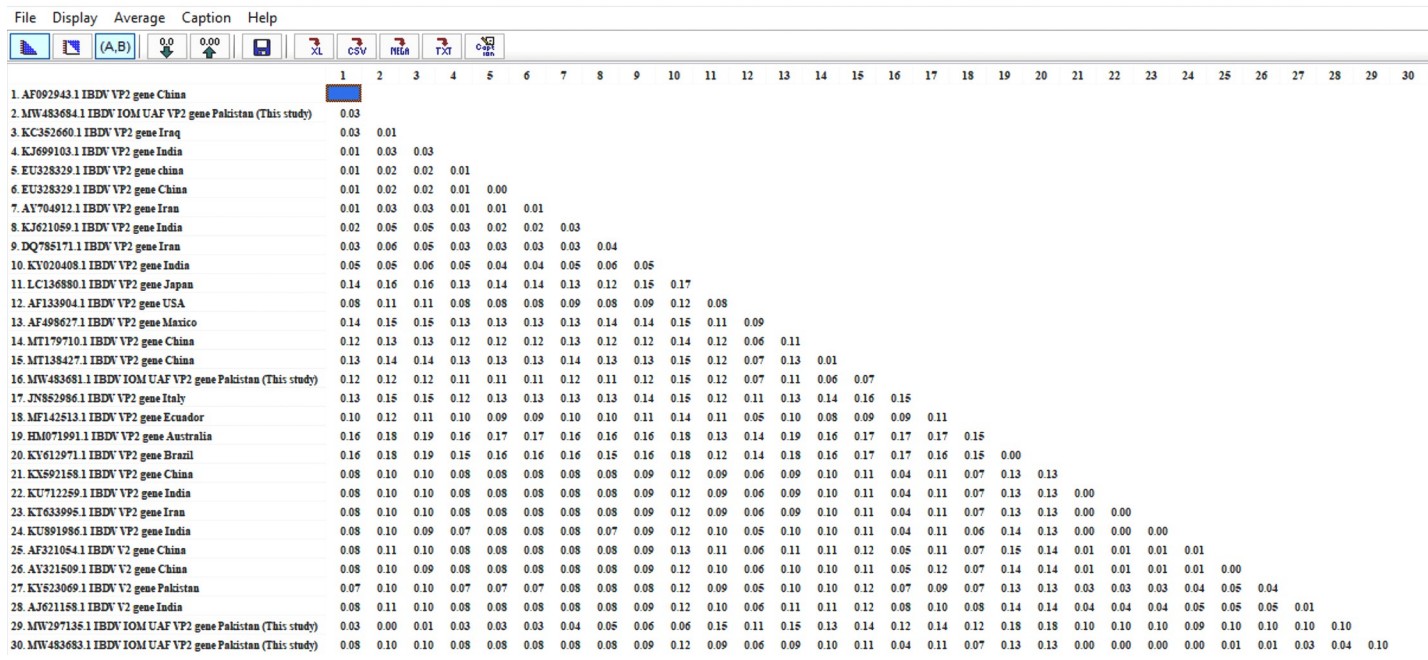

**Fig 5. Estimates of evolutionary divergence between sequences.**

## Acknowledgments

We are thankful to Dr. Mudasser Habib from NIAB for helping throughout the study and Rai Shafqat from Livestock department for the photographic work.

## Author Contributions

**Data curation:** Sanaullah Sajid, Amor Hedfi, Mohammed Almalki.

**Formal analysis:** Zia ud Din Sindhu, Mohammed Almalki.

**Investigation:** Mashkoor Mohsin Gilani, Amor Hedfi.

**Methodology:** Sanaullah Sajid, Sajjad ur Rahman, Mashkoor Mohsin Gilani, Shahid Mahmood.

**Supervision:** Sajjad ur Rahman.

**Validation:** Sanaullah Sajid, Shahid Mahmood.

**Writing – original draft:** Sanaullah Sajid.

**Writing – review & editing:** Zia ud Din Sindhu, Manel Ben Ali.

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
