## [Decision Letter · Decision Letter 0]

7 Jun 2021

PONE-D-21-08105

Molecular characterization and demographic study on infectious bursal disease virus in district Faisalabad

PLOS ONE

Dear Dr. Sajid,

Thank you for submitting your manuscript to PLOS ONE. After careful consideration, we feel that it has merit but does not fully meet PLOS ONE’s publication criteria as it currently stands. Therefore, we invite you to submit a revised version of the manuscript that addresses the points raised during the review process.

We look forward to receiving your revised manuscript.

Kind regards,

Sagheer Atta, Ph.D

Academic Editor

PLOS ONE

Journal Requirements:

"NO"

4. Please amend the manuscript submission data (via Edit Submission) to include authors Sajjad ur Rahman, Mashkoor Mohsin Gilani, Zia ud Din Sindhu.

5. Please ensure that you refer to Figures 1, 2, 3 and 5 in your text as, if accepted, production will need this reference to link the reader to the figures.

Reviewers' comments:

Reviewer's Responses to Questions

**Comments to the Author**

1. Is the manuscript technically sound, and do the data support the conclusions?

Reviewer #1: Yes

Reviewer #2: Yes

Reviewer #3: Yes

2. Has the statistical analysis been performed appropriately and rigorously? 

Reviewer #1: Yes

Reviewer #2: Yes

Reviewer #3: Yes

3. Have the authors made all data underlying the findings in their manuscript fully available?

Reviewer #1: Yes

Reviewer #2: Yes

Reviewer #3: Yes

4. Is the manuscript presented in an intelligible fashion and written in standard English?

Reviewer #1: Yes

Reviewer #2: Yes

Reviewer #3: Yes

5. Review Comments to the Author

Reviewer #1: The manuscript theme is good and work is done in beautiful manner, however there is a need of careful reading to improve the results section , introduction, technical name should be italic a little English spelling mistakes should be removed

Reviewer #2: Good study sound analysis need just careful reading and to remove the English grammar mistakes and technical name should be removed and a little improvement needed in the methodology section. there is need to conclude in a specific manner

Reviewer #3: This study looks very interesting to me. I have thoroughly reviewed your manuscripts and found some drawbacks which are important to revise it. I have suggested some amendments for this study. Authors should revise it carefully.

6. PLOS authors have the option to publish the peer review history of their article (what does this mean?). If published, this will include your full peer review and any attached files.

Reviewer #1: No

Reviewer #2: No

Reviewer #3: No

---

## [Author Response · Author response to Decision Letter 0]

25 Jun 2021

Respected sir, All the suggested corrections are made in the manuscript and the copy of approved manuscript is attached for further process. Thank you for your valuable comments.

---

## [Editor Report · Decision Letter 1]

30 Jun 2021

Molecular characterization and demographic study on infectious bursal disease virus in district Faisalabad

PONE-D-21-08105R1

Dear Dr. Sajid,

We’re pleased to inform you that your manuscript has been judged scientifically suitable for publication and will be formally accepted for publication once it meets all outstanding technical requirements.

Kind regards,

Sagheer Atta, Ph.D

Academic Editor

PLOS ONE
---

## [Editor Report · Acceptance letter]

5 Aug 2021

PONE-D-21-08105R1 

Molecular Characterization and Demographic Study on Infectious Bursal Disease Virus in Faisalabad District 

Dear Dr. Sajid:

I'm pleased to inform you that your manuscript has been deemed suitable for publication in PLOS ONE. Congratulations! Your manuscript is now with our production department. 

Kind regards, 

on behalf of

Dr. Sagheer Atta 

Academic Editor

PLOS ONE